# Short Course Antibiotic Therapy for Catheter-Related Septic Thrombosis: “Caveat Emptor!”: Duration of Therapy Should Not Be Set *a Priori*

**DOI:** 10.3390/pathogens13070529

**Published:** 2024-06-22

**Authors:** Alberto Enrico Maraolo, Giancarlo Ceccarelli, Mario Venditti, Alessandra Oliva

**Affiliations:** 1Section of Infectious Diseases, Department of Clinical Medicine and Surgery, University of Naples Federico II, 80131 Naples, Italy; albertomaraolo@mail.com; 2Department of Public Health and Infectious Diseases, Sapienza University of Rome, Piazzale Aldo Moro 5, 00185 Rome, Italy; giancarlo.ceccarelli@uniroma1.it (G.C.); mario.venditti@uniroma1.it (M.V.); 3Infectious Diseases Department, Azienda Ospedaliero Universitaria Policlinico Umberto I, 00161 Rome, Italy

**Keywords:** septic thrombosis (ST), catheter-related bloodstream infections (CRBSIs), follow-up blood culture (FUBC), treatment duration

## Abstract

There is a growing body of evidence showing no significant difference in clinical outcomes in patients with uncomplicated Gram-negative bloodstream infections (BSIs) receiving 7 or 14 days of therapy. However, the scenario may differ when complicated forms of BSI, such as catheter-related BSIs (CRBSIs) burdened by septic thrombosis (ST), are considered. A recent study showed that a short course of antimicrobial therapy (≤3 weeks) had similar outcomes to a prolonged course on CRBSI-ST. From this perspective, starting from the desirable goal of shortening the treatment duration, we discuss how the path to the correct diagnosis and management of CRBSI-ST may be paved with several challenges. Indeed, patients with ST due to Gram-negative bacteria display prolonged bacteremia despite an indolent clinical course, requiring an extended course of antibiotic treatment guided by negative FUBCs results, which should be considered the real driver of the decision-making process establishing the length of antibiotic therapy in CRBSI-ST. Shortening treatment of complicated CRBSIs burdened by ST is ambitious and advisable; however, a dynamic and tailored approach driven by a tangible outcome such as negative FUBCs rather than a fixed-duration paradigm should be implemented for the optimal antimicrobial duration.

## 1. Introduction

One of the recent trends in the field of antimicrobial therapy is to administer anti-infective agents for the shortest possible duration, based on the notable principle that “shorter is better”, aimed at reducing unnecessary exposure to preserve drugs, minimize toxicity and selection of resistant pathogens as well as to contain costs [1]. Bloodstream infections (BSIs) represent no exception, and for instance, there is a growing body of evidence showing no significant difference in clinical outcomes comparing patients with uncomplicated Gram-negative BSI allocated to receive 7 days and the ones assigned to receive 14 days in randomized clinical trials (RCTs) [2]. Even in uncomplicated candidemia some authors advocated a shorter course than the usual 14 days after the first negative follow-up blood culture (FUBC) relying on a retrospective study [3]. From analogous evidence, other researchers suggested that in uncomplicated *Staphyloccocus aureus* BSI, namely the paradigmatic form of bacteremia with metastatic potential, less than 14 days of therapy may suffice according to real-world evidence (RWE) data [4].

Actually, there is often a discrepancy between the results from RCTs and from RWE. In an elegant review collating and discussing observational studies and RCTs conceived to address different antibiotic treatment durations, with a focus on Gram-negative BSI as a case study, the authors succeeded in showing that the two diverse types of study can generate opposite or conflicting findings [5]. Against the backdrop of a Bayesian framework, the authors showed how the posterior probability of better outcomes associated with longer courses of antibiotic therapy when considering mortality as the endpoint was 91% from observational studies and less than half (42%) from RCTs [5]. On the contrary, with regard to infection recurrence, the posterior probability that longer courses were better was 53% for the RCTs and 12% for the observational studies [5].

These conflicting findings can be easily explained by the several distortions that can affect observational studies: among the many, the ones that stand out are confounding by indication or severity, ascertainment bias and immortal time bias [6]. These important issues can be at least partially mitigated by resorting to proper statistical designs, including for instance the use of propensity score (PS) to balance the distribution of observed (but not the unknown) baseline covariates between different treatment groups [7], and the use of landmark analysis to attenuate immortal time bias; although, no method completely accounts for it [8].

As is well known, results from observational studies should always be interpreted judiciously, but considerable caution is imperative when different lengths of therapy are compared due to the above-mentioned issues. That applies to “uncomplicated” BSI and even more to their “complicated” counterparts.

As a matter of fact, the common denominator of the studies whose results were briefly in the preamble is the key adjective “uncomplicated”: both RCTs and observational studies showing no higher mortality in short-treatment groups typically address patients whose baseline features imply a good prognosis from the start. Indeed, in the case of uncomplicated Gram-negative BSI, there is a concordance between findings from RCTs [2] and from a large observational study whose data were used to emulate a target trial [9].

The scenario is completely different when “complicated” forms of BSI are at play. In this respect, catheter-related bloodstream infections (CRBSIs) complicated by septic thrombosis (ST) represent one the most complex condition in this setting whose management has been historically rooted in three tenets: prompt catheter withdrawal, prolonged antibiotic treatment and tailored assessment of the need for anticoagulation [10].

The aim of this short review is to illustrate the issues linked with the interpretation of findings from observational studies addressing the duration of therapy of complicated BSIs, by resorting to a recently published retrospective experience as a case study [11]. We are worried that the message conveyed by this study, run in a peculiar population that is difficult to enroll in properly conducted RCTs and for which real-word evidence is important, may be misinterpreted as green-lighting a therapeutic approach of short antimicrobial duration that is notable in theory, but likely not always feasible in practice. This work may serve to underline how the path to the correct diagnosis and management of complicated BSI in general and of CRBSI-ST in particular is paved with several pitfalls.

## 2. Shortening Duration of Catheter-Related Bloodstream Infections (CRBSIs) Complicated by Septic Thrombosis (ST): The Case Study

In a very recent single-center and retrospective experience focused on patients with CRBSI-ST, French researchers compared a short course of antimicrobial therapy (no more than 3 weeks) with a prolonged course, founding no significant difference in the composite outcome of 30-day post-treatment all-cause mortality or relapse [11].

The former approach was adopted in France after a national consensus paper advocating a shorter treatment of 21 days for ST based on the lack of data in favor of longer courses [12]. This is in opposition to United States 2009 guidelines, recommending a minimum of 3 or 4 weeks of antibiotic therapy (up to 6), although relying on expert opinion as well [13].

The French authors included patients showing a combination of deep-vein thrombosis (DVT) homolateral to the catheter and certain or possible CRBSI. It should be noted that as reported, the risk of the primary outcome was similar among groups, but actually, the risk was more than two-fold higher in patients undergoing shorter treatment, although not in a statistically significant fashion: hazard ratio (after inverse probability of treatment weighting) equal to 2.16, 95% confidence interval 0.68–6.88.

In our opinion, interpretation of study results is problematic in many ways. Specifically, we will delve into its limitations concerning several aspects.

The first limitation is the proper definition of the clinical problem, since the risk of misclassifying patients threatens the study’s results and its generalizability from the start.

Second, the factors driving the choice of diverse treatment durations and the way to assess the response to treatment are limiting, since choosing a fixed number of antimicrobial days is a risky strategy in case of complicated infections.

Third, the statistical methods employed to analyze data are limiting, because even the use of some techniques in a causal framework does not fully amend the inherent weaknesses of an observation retrospective study, and the misuse of these techniques can further jeopardize the validity of such study.

## 3. Catheter-Related Bloodstream Infections (CRBSIs) Complicated by Septic Thrombosis (ST): Beware the Right Definition and the Appropriate Diagnosis

Distinguishing between complicated and uncomplicated forms of BSI is crucial.

As for *S. aureus* infections, likely the most investigated among BSI instances, a seminal definition of the complicated form was provided by Fowler and colleagues as early as 2003, based on criteria such as infection-related mortality, embolic stroke, metastatic or locally complicated or recurrent infection [14]. Of course, the problem of this kind of definition is that it relies on outcomes that are unknown at the beginning of the clinical course, therefore subsequent definitions attempted to focus more on risk factors, as done by the Infectious Diseases Society of America (IDSA) guidelines in 2011: according to the IDSA document, the absence of endocarditis and of implanted prostheses, along with other elements, allows us to categorize a BSI as not complicated [15]. A bundle of actions to differentiate among the two forms, that imply a distinct management, is commonly accepted [16], but even for *S. aureus*, the criteria to classify a BSI episode as complicated or uncomplicated are based on low-quality evidence and areas of controversy are present [17].

As for candidemia, adherence to international recommendations outlined in the EQUAL *Candida* score, representing a management bundle, is an independent predictor of survival [18]. Nevertheless, the boundary between complicated and uncomplicated forms is somewhat porous [3]; although, the absence of metastatic foci and of deep-seated candidiasis may be considered as a marker of a not-complicated infection [19].

Eventually, as for Gram-negative BSI, there is no universally accepted set of criteria to define complicated infection, as exemplified by the heterogenous inclusion and exclusion criteria of the RCTs [20,21,22], informing the individual patient data meta-analysis whose results show equivalent outcomes between short-course and long-course treatments for uncomplicated BSI [2]. Some unifying tracts were the exclusion of patients affected by immunosuppression, polymicrobial infection, endocarditis or other infections requiring prolonged treatment according to international consensus [20,21,22].

At any rate, whichever the causative pathogen, ST is a complication-defining condition in patients with CRBSI. While septic thrombophlebitis properly speaking involves direct invasion of the vascular wall by pathogens from a contiguous focus of infection, resulting in vein inflammation and thrombosis, and, potentially, in secondary bacteremia, the term septic thrombosis involves inflammation and bacterial colonization of the thrombus by a hematogenous seeding from distant foci of infection [14].

Nevertheless, case definition of ST linked with CRBSI is very subtle because it is not straightforward to differentiate between the concurrent presence of catheter-related thrombosis and bacteremia as well as the actual suppuration of the thrombus responsible for ST, owing to the two-way association between bacteremia, the intravascular device and the development of venous thrombosis [23]. Diagnosis of this complication implies persistent bacteremia after at least 72 h of appropriate antimicrobial therapy, thus prompting screening for the occult source of bacterial dissemination in the bloodstream, ideally following the rapid removal or exchange of the device within 48 h of the onset of bacteremia [23]. Otherwise, there is a high risk of misclassification bias, unclear in this case whether differential or non-differential, by defining as ST cases of sterile DVT.

Another point worth mentioning is the use of imaging. In the French study, the authors declared that they retrospectively screened patients using the hospital’s radiology software to identify every upper limb DVT defined as demonstration of a thrombus by ultrasonography (US) in a deep vein of the upper limb. Anyway, the rate of missed diagnosis of upper-limb DVT by ultrasound was higher than with computed tomography (CT) scan; in fact, when US is inconclusive, CT venography is often used to confirm or exclude DVT [24]. As previously reported in our experience, persistent GNB bacteremia can be related to undiagnosed underlying thrombophlebitis, especially in polytraumatized or critically ill patients heavily assisted with vascular devices [25,26,27]. This may have introduced another selection bias, excluding from the analysis all cases not detected by US.

## 4. Catheter-Related Bloodstream Infections (CRBSIs) Complicated by Septic Thrombosis (ST): How to Monitor Response to Treatment

Performing follow-up blood cultures (FUBCs) represents a key strategy to assess the microbiological response under appropriate antimicrobial treatment in cases of infections by *S. aureus*, *Enterococcus* spp. and in case of Gram-negative BSI when severe and/or high-inoculum infections, non-eradicable foci or immunosuppressed patients [28]. Documentation of microbiological clearance is crucial in the case of candidemia to establish treatment duration [29]. For sure, the indiscriminate use of FUBCs in uncomplicated Gram-negative BSI does not seem to exert a beneficial effect, on the contrary, potentially driving an increase in the length of stay [30], but in the presence of complicated forms the achievement of microbiological eradication should be determined to appropriately monitor response.

In line with the French recommendations that offer no guidance in this respect [12], in the study under scrutiny, there is no information about the impact of important variables on duration choice such as infections by a resistant organism and, crucially, about number, timing and results of FUBCs. Antibiotic duration in the short-course group was equal to 2 weeks or less in one-third of cases, being the median of 16 days, so violating the French protocol itself, and it is not clear what dictated the duration on a case-by-case basis [11].

Indeed, no definition was provided about the strategy used to declare clinical improvement or healing of patients. Being a retrospective study, we assume that the criterion used to define the resolution of ST was the duration of antibiotic treatment in the absence of documented clinical relapse, but we believe that demonstration of negative FUBCs results should be the real driver of the decision-making process establishing the length of antibiotic therapy.

In previous studies, we demonstrated that patients with septic thrombophlebitis due to Gram-negative bacteria (GNB) displayed prolonged bacteremia with significant challenges in pathogen clearance despite a mild and indolent clinical course, often characterized by the absence of fever, hemodynamic stability, and low serum levels of procalcitonin (Figure 1). In particular, the swift decline in PCT levels, followed by sustained normalization of serum concentration despite the ongoing presence of bacteremia, suggests a potential mechanism of immune tolerance. This involves the selective suppression of specific pro-inflammatory pathways activated by bacterial endotoxins or cytokines, which could hinder PCT production and cause a prolonged clinically benign course despite the failure to eliminate the microorganism [30].

In these cases, an extended course of antibiotic treatment was necessary to clear the septic focus, even if patients seemingly displayed no clinical signs of bloodstream infection [24,25,26,27]. Notably, in the French study GNB accounted for only 15.1% of the cases overall, but endovascular infections in general, whichever the causative agent, require FUBCs to demonstrate microbiological clearance, which is a watershed moment in the course of the disease to establish the treatment duration [31,32,33].

## 5. The Limitations of Observational Studies Aimed at Informing Antibiotic Treatment Duration

In the French study, there is a commendable attempt to mitigate the several biases afflicting observational studies in several ways, firstly developing a PS model to balance baseline prognostic covariates [11]. Nonetheless, there are various critical aspects deserving comments, offering guidance to avoid mistakes or misinterpretations in future similar studies.

For instance, the PS model constructed was at risk of misspecification due to some issues regarding variable selection. The PS model should not be parsimonious, including as many potential confounders as possible based on subject matter knowledge [34]. A selection criterion based on prediction of the exposure (choice of treatment duration) may miss variables related only to the outcome, that should be included differently from the ones associated only with the exposure, so p-value-based methods are to be avoided [34]. As a matter of fact, the model did not include key variables such as Pitt bacteremia score, used for more than three decades to predict mortality in the case of BSI and more specific than scores linked with sepsis [35].

To attenuate immortal-time bias, authors used a landmark method, excluding patients classified as early deaths: in detail, “death within 21 days of the treatment initiation or on the day treatment ended”. Therefore, it is not clear whether the landmark was set at the end of the period of short-course treatment, to ensure no deaths occurred in the short treatment group, or excluding all patients who died during therapy in both groups: according to the different method, the bias shifts against one or the other therapeutic approach [5].

A further problem was the choice of a composite outcome including both death and a non-fatal event (relapse): appropriate analysis is controversial, but the adopted practice of analyzing time to the first event through a proportional hazards model has been strongly criticized for ignoring the dissimilar relevance between component events and for underutilization of outcome data, since recurrent-event data may be left unused [36]. Additionally, when the event rate of a component of a composite outcome is not trivial (at least 5%), a competing risk analysis should be performed to better understand the treatment effect on the component [37]. In the prolonged-course group, the composite endpoint was experienced in 14.4% among both unweighted and weighted populations, thus at least one component is over 5%. Moreover, analyzing as separated secondary outcomes 90-day all-cause mortality and 90-day relapse outside a competing risk analysis framework is not correct.

## 6. Proposal for a Practical Approach to the Management of ST

Patients at risk of developing ST include those who are critically ill, hospitalized in the ICU, and/or have a central line in place. Additionally, polytrauma is a significant risk factor for GN-ST. Notably, most patients with polytrauma develop lower extremity venous thrombosis near the site of a bone fracture [27] (Table 1).

For patients at risk for ST, a diagnostic work-up is essential in order to confirm or exclude the presence of this infectious condition (Figure 2). Indeed, the suspicion is primarily driven by laboratory-confirmed persistent GN bacteremia, defined as repeatedly positive blood cultures (BC) after at least 96 h of appropriate antibiotic treatment and at least 48 h after the removal of all potentially infected endovascular devices. If persistent bacteremia is detected, along with venous thrombosis in at least one anatomical site (e.g., supra-aortic trunks, upper or lower limbs, abdomen) confirmed by Doppler ultrasound and/or CTA, and the clinical course is consistent with ST, while other recognized sources of infection are excluded, the diagnosis of ST is confirmed (Figure 2).

As for imaging methods, CTA should be preferred over Doppler ultrasound in the case of polytrauma involving lower extremity or if Doppler ultrasound results are inconclusive, and there is a high suspicion of ST (Table 1).

Once diagnosed, the therapeutic approach to ST should be multidisciplinary, involving vascular surgeons, cardiologists, intensivists for appropriate anticoagulation, and infectious disease specialists. As a matter of fact, the mainstay of therapeutic management for ST involves prolonged use of in vitro active antimicrobials, either alone or in combination depending on the offending microorganism and its resistance profile [38], together with appropriate anticoagulation (Table 1).

Monitoring should involve repeating follow-up blood cultures (FUBCs) every 24–48 h following device removal until negative FUBC results are obtained. It is important to note that the absence of fever and/or normalization of procalcitonin (PCT) levels do not definitively exclude the possibility of persistent GN bacteremia (Figure 2). Indeed, only negative FUBCs, rather than clinical improvement, should be considered the true indicator of ST resolution. Proposed recommendations for or against FUBC samplings are depicted in Figure 2.

## 7. Conclusions

In conclusion, shortening the treatment of complicated CRBSIs burdened by ST is an ambitious and desirable goal from an antimicrobial stewardship perspective, but it needs to be shored up by strong evidence, possibly from randomized trials, recruiting patients fulfilling stringent diagnostic criteria. More than a fixed-duration paradigm, a more dynamic approach should be implemented, individually tailoring and fine-tuning antimicrobial duration according to a tangible outcome such as negative FUBCs results.

## Figures and Tables

**Figure 1 pathogens-13-00529-f001:**
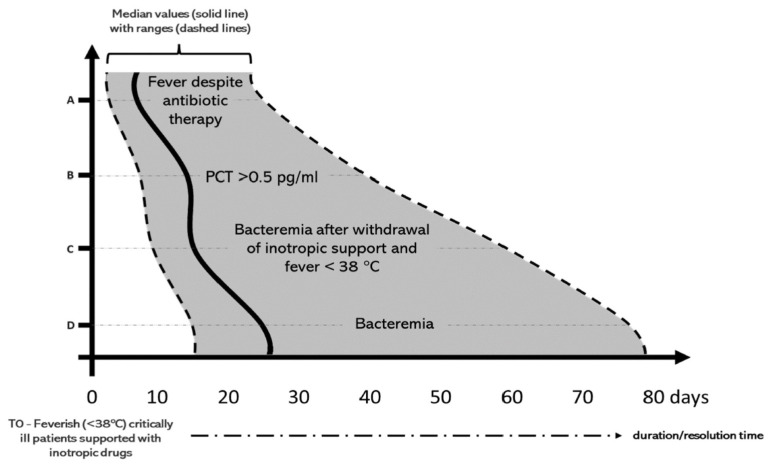
Comparison between the median times (solid line) with ranges (dashed lines) of duration of the 4 clinical, laboratory and microbiological prominent features of Gram-negative septic thrombosis: (A) fever, (B) procalcitonin (PCT), (C) persistent bacteremia after clinical improvement defined as withdrawal of inotropic support and reduction of fever under 38 °C, (D) global duration of bacteremia (based on [26]).

**Figure 2 pathogens-13-00529-f002:**
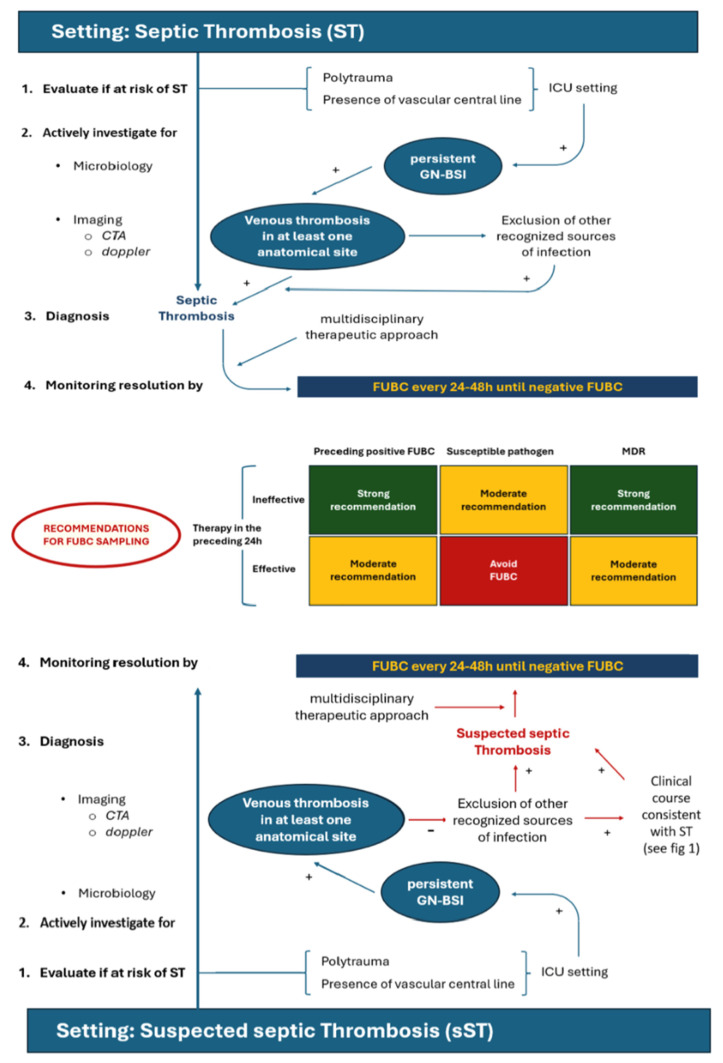
Diagnostic and therapeutic algorithm for the management of Gram-negative (GN) septic thrombosis. ICU: Intensive Care Unit; CTA: computed tomography angiography; GN-BSI: Gram-negative bloodstream infection; FUBC: follow-up blood cultures; MDR: multi-drug resistance. Green: strong recommendation; Yellow: Moderate recommendation; Red: Avoid FUBC.

**Table 1 pathogens-13-00529-t001:** Proposed practical diagnostic and therapeutic approach to Gram-negative septic thrombosis. ICU: Intensive Care Unit; GN: Gram-negative; BC: blood culture; CTA: computed tomography angiography; ST: septic thrombosis; FUBC: follow-up blood cultures.

Conditions at Risk	Diagnosis	Imaging	Treatment	Monitoring
ICU setting Polytrauma Presence of vascular central line	Laboratory-confirmed persistent GN bacteremia (defined as repeatedly positive BC after at least 96h of appropriate antibiotic treatment and at least 48h since the removal of all potentially infected endovascular devices) + Exclusion of other recognized sources of infection + Venous thrombosis in at least one anatomical site (i.e., supra-aortic trunks, upper or lower limbs, abdomen) assessed by Doppler ultrasound and/or CTA	CTA or Doppler ultrasound Prefer CTA scan if polytrauma involving lower extremity or if Doppler ultrasound inconclusive and high suspicion of ST	Active in vitro antimicrobials (alone or in combination depending on the offending microorganism) + Anticoagulation Vascular surgeons/cardiologists/ intensivists consultation for appropriate anticoagulation	FUBC every 24–48 h following device removal until negative FUBC Note: The absence of fever and/or a return of PCT to normal values do not constitute criteria for definitively excluding the possibility of persistent GN bacteremia.

## Data Availability

No new data were created or analyzed in this study. Data sharing is not applicable to this article.

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
