# Peer review of "Short Course Antibiotic Therapy for Catheter-Related Septic Thrombosis: “Caveat Emptor!”: Duration of Therapy Should Not Be Set a Priori"

_pathogens, 2024, doi:10.3390/pathogens13070529_

Round 1

Reviewer 1 Report

Comments and Suggestions for Authors

The present review deals with the issue of catheter-related septic thrombosis. This clinical entity has controversial definition; however, the suggested treatment duration is at least 3 weeks. I would like to commend the authors for their excellent report on this often neglected topic. Please, consider my comments before publication.

1)      The authors define septic thrombosis as persistent bacteremia. Of course, no one can tell that a thrombus homolateral to the catheter is infected. I reckon that the French paper used a more practical approach by considering the worst case scenario than providing patients briefer regimens. Meanwhile, the issue of repeat blood cultures after treatment initiation is not settled, particularly for the Gram-negative bacteria. Thus, many would not repeat blood cultures at a routine basis.

2)      The infection severity status can be evaluated through a score, such as Pitt Bacteremia score or the SOFA score. Indeed, the French group did not include a severity index in the propensity score modeling.

3)      An occult infection locus can be discovered via a variety of imaging tools, it can be non-vascular, or vascular. A repeat blood culture in such case is part of a diagnostic work-up.

4)      Finally, a brief paragraph about the authors’ experience and their practical routine would be a welcome addition to the report. Do the authors consider ultrasound in every patient? Do they perform blood sampling on 48 or 72 hours after treatment initiation, or even later? Do they have a practical algorithm for this infection?

Author Response

We thank the reviewer for their valuable insights and comments, which led us to improve the manuscript. We followed reviewers’ comments and we hope that in this revised version the manuscript will be suitable for publication. We also added some references, a table and a Figure.

Reviewer 1

The present review deals with the issue of catheter-related septic thrombosis. This clinical entity has controversial definition; however, the suggested treatment duration is at least 3 weeks. I would like to commend the authors for their excellent report on this often neglected topic. Please, consider my comments before publication.

1) The authors define septic thrombosis as persistent bacteremia. Of course, no one can tell that a thrombus homolateral to the catheter is infected. I reckon that the French paper used a more practical approach by considering the worst case scenario than providing patients briefer regimens. Meanwhile, the issue of repeat blood cultures after treatment initiation is not settled, particularly for the Gram-negative bacteria. Thus, many would not repeat blood cultures at a routine basis.
A: Thank you for the comment. We added a figure summarizing the conditions where to perform FUBC in GN bacteremia.

2) The infection severity status can be evaluated through a score, such as Pitt Bacteremia score or the SOFA score. Indeed, the French group did not include a severity index in the propensity score modeling.
A: Thank you. We added the following sentence: As matter of fact the model did not include key variables such as Pitt bacteremia score, used for more than 3 decades to predict mortality in case of BSI and more specific than scores linked with sepsis [35].

3) An occult infection locus can be discovered via a variety of imaging tools, it can be non-vascular, or vascular. A repeat blood culture in such case is part of a diagnostic work-up.
A: discussion on the role of imaging tools is present (lines 159-168)

4) Finally, a brief paragraph about the authors’ experience and their practical routine would be a welcome addition to the report. Do the authors consider ultrasound in every patient? Do they perform blood sampling on 48 or 72 hours
after treatment initiation, or even later? Do they have a practical algorithm for this infection?
Thank you for your sugges/on. We added a short paragraph (en/tled “Proposal for a prac/cal approach to the management of ST”) concerning our prac/cal approach to confirm or exclude the presence of ST, along with corresponding Figure and Table.

Reviewer 2 Report

Comments and Suggestions for Authors

This manuscript provides an analysis of short course antibiotic therapy in the context of a published manuscript that authors use as a case study to introduce discussion. Authors present excellent analysis of this previous publication, but I find the order of information very difficult to follow. Perhaps it could be restructured to improve readability? Specific suggestions below.

-As written, this manuscript reads more like a journal club of a single article versus a review. If that is what was intended, it would be beneficial to have an introduction, then a section specifically devoted to (briefly) summarizing the findings/points you wish to address. For example, you could include lines 51-55, 61-65, 73-74, etc. in this summary. 

-If this intention was to truly do a review, it may be helpful to add additional references to support or refute points as appropriate. 

-The order is a bit confusing to me. Not all information within each section seems to align with the title. Perhaps you could order it something like:

1. Introduction: cover the trend toward shorter courses, then go into problems with assessing durations by observational trial (currently lines 134-143), then discuss how you will use one study as a case example (currently lines 66-70). 

2. Case/publication summary as described above

3. Definition/diagnosis: could discuss here uncomplicated vs. complicated (currently lines 44-51), DVT vs. ST (lines 73-83), imaging (85-94). 

4. Follow-up: currently section 3). As written, this section is confusing. Perhaps start with what follow-up is recommended in the literature for CRBSI with ST, then move into what this study did, then move into your concerns about that? 

5. Limitations (current section 4): I think it would be helpful to introduce this earlier, then you could still keep your analysis of their limitations in this section. As written, the paragraphs seem a little out of order or like they should go in another section. Should lines 159-169 go in the monitoring/follow-up section instead of the limitations portion? It doesn't seem to fit where it is. 

6. Could consider a table for definitions/diagnosis as currently recommended and use that as your intro to the definitions/diagnosis portion as additional background for your readers.  

Comments on the Quality of English Language

No specific comments on English language. 

Author Response

Reviewer2
This manuscript provides an analysis of short course antibiotic therapy in the context of a published manuscript that authors use as a case study to introduce discussion. Authors present excellent analysis of this previous publication, but I find the order of information very difficult to follow. Perhaps it could be restructured to improve readability? Specific suggestions below.

-As written, this manuscript reads more like a journal club of a single article versus a review. If that is what was intended, it would be beneficial to have an introduction, then a section specifically devoted to (briefly) summarizing the findings/points you wish to address. For example, you could include lines 51-55, 61-65, 73-74, etc. in this summary.
A: done

-If this intention was to truly do a review, it may be helpful to add additional references to support or refute points as appropriate.
A: we added additional references

-The order is a bit confusing to me. Not all information within each section seems to align with the title. Perhaps you could order it something like:
1. Introduction: cover the trend toward shorter courses, then go into problems with assessing durations by observational trial (currently lines 134-143), then discuss how you will use one study as a case example (currently lines 66-70).
2. Case/publication summary as described above
3. Definition/diagnosis: could discuss here uncomplicated vs. complicated (currently lines 44-51), DVT vs. ST (lines 73-83), imaging (85-94).
4. Follow-up: currently section 3). As written, this section is confusing. Perhaps start with what follow-up is recommended in the literature for CRBSI with ST, then move into what this study did, then move into your concerns about that?
5. Limitations (current section 4): I think it would be helpful to introduce this earlier, then you could still keep your analysis of their limitations in this section. As written, the paragraphs seem a little out of order or like they should go in another section. Should lines 159-169 go in the monitoring/follow-up section instead of the limitations portion? It doesn't seem to fit where it is.
A: according to reviewer’s suggestion, we re-ordered the points as follows: 1 Introduction; 2 The case study; 3 Definition and diagnosis; 4 monitor of response to treatment; 5 limitation; 6 conclusion.
6. Could consider a table for definitions/diagnosis as currently recommended and use that as your intro to the definitions/diagnosis portion as additional background for your readers.

Thank you for your sugges/on. Following the reviewer’s #1 comments, we added a short paragraph (en/tled “Proposal for a prac/cal approach to the management of ST”) concerning our prac/cal approach to confirm or exclude the presence of ST, along with corresponding Figure and Table.

Round 2

Reviewer 2 Report

Comments and Suggestions for Authors

Authors have made significant revisions and added many references, both of which improve readability and help drive home their points. No further comments.